# Enhancement of Antimicrobial and Antiproliferative Activities of Standardized Frankincense Extract Using Optimized Self-Nanoemulsifying Delivery System

**Shereen S. El-Mancy** [1]**, Alaadin E. El-Haddad** [2,*]**, Walaa A. Alshareef** [3]**, Amr M. Saadeldeen** [4]**, Soad Z. El-Emam** [5] **and Osama S. Elnahas** [1]

[1] Department of Pharmaceutics and Industrial Pharmacy, Faculty of Pharmacy, October 6 University, Giza 12585, Egypt; shereenelmancy@o6u.edu.eg (S.S.E.-M.); osamaelnahas@o6u.edu.eg (O.S.E.)

[2] Department of Pharmacognosy, Faculty of Pharmacy, October 6 University, Giza 12585, Egypt

[3] Department of Microbiology and Immunology, Faculty of Pharmacy, October 6 University, Giza 12585, Egypt; dr.walaa@o6u.edu.eg

[4] Department of Pharmacognosy, School of Pharmacy, Newgiza University, Giza 12556, Egypt; amr.saadeldeen@ngu.edu.eg

[5] Department of Pharmacology and Toxicology, Faculty of Pharmacy, October 6 University, Giza 12585, Egypt; soadzakaria@o6u.edu.eg

* Correspondence: alaa_elhaddad.ph@o6u.edu.eg or Haddad002@yahoo.com; Fax: +2-0238354275

**Abstract:** Boswellic acids (BAs) are the main bioactive compounds of frankincense, a natural resin obtained from the genus *Boswellia*. This study aimed to develop a self-nanoemulsifying delivery system (SNEDS) to improve the antimicrobial and antiproliferative activities of standardized frankincense extract (Fr-extract). Fr-extract was standardized, and BA content was quantified using the developed HPLC-UV method. Screening studies of excipients followed by formula optimization using a mixture simplex lattice design was employed. The optimized Fr-SENDS formulation was characterized. Furthermore, microbiological and antiproliferative assessments of the standardized Fr-extract and Fr-SNEDS were evaluated. Quantification demonstrated that the major constituent is 11-keto-boswellic acid (KBA) (16.25%) among BA content (44.96%). The optimized Fr-SENDS (composed of 5% Capryol™ 90, 48.7% Gelucire® 44/14 and 46.3% ethanol) showed spherical nanosized dispersions with DS, PDI, and zeta potential of 17.9 nm, 0.2, and −14.5 mV, respectively. Fr-SNEDS exhibited lower MIC and MBC values compared with Fr-extract against pathogens conjugated with lung cancer and was comparable to reference antimicrobials. Fr-SNEDS showed superior antiproliferative activity over Fr-extract, with IC$_{50}$ values of 20.49 and 109.5 µg mL$^{-1}$, respectively. In conclusion, the optimized Fr-SNEDS could be easily developed and manufactured at a low cost and the in vitro results support its use as a potential adjuvant oral therapy for lung cancer. Further in vivo studies could be continued to assess the therapeutic efficiency of the prepared system.

**Keywords:** antiproliferative; boswellic acids; frankincense; MRSA; self-nanoemulsifying delivery system; simplex lattice design

## 1. Introduction

Nowadays, researchers are moving towards the utilization of phytoconstituents, owing to their therapeutic effectiveness with low side effects [1]. Frankincense, or olibanum, is a natural resin obtained from trees of the genus *Boswellia*, which is widespread in Arabia and India. Frankincense is comprised mainly of oleogum and terpenoids. Frankincense pentacyclic triterpenes are considered to be the most bioactive. Among triterpenoids, boswellic acids (BAs) are of particular interest, particularly *β*-boswellic acid and 11-Keto-*β*-boswellic acid (KBA) and their 3-*O*-acetyl esters (AKBAs) [1]. The major composition of

the *Boswellia* resin shows approximately 50–60% of various BAs, and in herbal supplements, up to 65% [2]. The analysis of these triterpenes is performed by different analytical methods based on HPLC-DAD [3] and LC/MS [1,4]. A novel chromatographic separation method for selective analysis of the $\alpha$- and $\beta$-isomers of KBA and AKBA showed that $\alpha$-KBA produces only a small proportion (1.1–7.8%) of the total KBAs in oleogum resins of *B. sacra*. Similarly, $\alpha$-AKBA produces only 0.5–6.4% of the total AKBAs and the majors were $\beta$-isomers [5]. AKBA possesses antibacterial activity against Gram-positive bacteria and inhibits the biofilm generated by *Staphylococcus* [6]. On other hand, *Boswellia* triterpenoids were reported to have antitumor properties [7], apoptotic effects, and inhibited the protein synthesis in human leukemia cells [8]. Moreover, both $\alpha$- and $\beta$-KBA exhibited cytotoxicity against triple-negative breast cancer (TNBC) cell lines and induced in vivo apoptosis in MDA-MB-231 xenografts, where the $\beta$-isomers of KBA and AKBA demonstrated higher cytotoxic efficacies [5]. Furthermore, AKBA has an inhibitory effect on lipoxygenases following inhibition of cell proliferation [9].

AKBA is a class IV drug according to bio classification system (BCS) [10], and it is a challenge to improve its solubility and permeability. The development of nanocarriers offers many advantages, such as improving solubility, bioavailability, and stability, which consequently increases the therapeutic activity of drugs [11]. There are several studies that illustrated the use of nano-preparations as potential drug delivery systems for phytoconstituents [12–14]. Over recent years, the utilization of self-nanoemulsifying delivery systems (SNEDS) has been of great interest. After oral administration of SNEDS, and upon contact with GIT fluids, they produce nanoemulsions (droplet sizes 20–200 nm), which allow a large interfacial surface area for drug absorption, resulting in an enhanced oral bioavailability of poorly water-soluble drugs [15–17]. SNEDS have been used for developing drug delivery systems of some phytoconstituents such as curcumin [18], cinnamon oil [19], morin [13], curcumin, and thymoquinone [20]. Several studies have showed that SNEDS improved the anticancer activity of drugs [21,22] and of phytoconstituents [23,24]. Self-emulsification depends on the characters of oils, surfactants, and cosurfactants, their ratios, and the temperature during emulsification procedure. Moreover, efficient self-emulsifying systems need very specific combinations of pharmaceutical excipients [15,25,26]. Pharmaceutical optimization of SNEDS is often based on the construction of phase diagrams, which is time-consuming. Design of experiment (DoE) has been used for formulation development and optimization to save time and effort, where application of a suitable design allows screening of different variables simultaneously. The systematic optimization technique includes measuring the response variables, fitting a mathematical model, and conducting appropriate statistical tests to ensure the optimum formulation composition [25,27].

Herbal medicines were not considered for the development of novel dosage forms due to formulation difficulties. Modern formulation can be utilized to enhance phytoconstituents efficacy [28]. In this study, we present a rationale for the development of SNEDS as a suitable oral form of Fr-extract. BAs were quantified in Fr-extract; subsequently, Fr-SNEDS were prepared using DoE for formulation optimization. Finally, the optimized Fr-SNEDS formulation was subjected to antimicrobial and antiproliferative studies compared with Fr-extract.

## 2. Material and Methods

### 2.1. Materials

2.1.1. Chemicals

*Boswellia sacra* Flueck. gum oleoresin was purchased from Harraz Egyptian herbal store, Cairo, Egypt. KBA, AKBA, doxorubicin-hydrochloride (≥98.0%), MTT, propidium iodide, DMSO, Tween 80, isopropyl myristate (IPM), Kolliphor® RH40, and propylene glycol (PG) were purchased from Merck, Darmstadt, Germany. Dent (Oxoid *Helicobacter*

*pylori*) selective supplement, nitrofurantoin, amoxacillin/clavulanic acid, cefepime, metronidazole, and fluconazole discs were purchased from HiMedia Laboratories, Mumbai, India. Capryol™ 90, Labrafil® M 1944 CS, Gelucire® 44/14, Labrasol® were a kind gift from Gattefosse, Saint-Priest, France. Polyethylene glycol 400 (PEG400) was purchased from Fluka, Buchs, Switzerland. All other chemicals were of analytical reagent grade.

### 2.1.2. Microbial Strains and Cell Lines

Gram-positive microbial strains *Bacillus subttilis* (ATCC 6633), *Micrococcus luteus* (ATCC 27566), *Staphylococcus aureus* (ATCC 25923), *Staphylococcus epidermidis* (ATCC 12228), and MRSA (ATCC 43300), along with Gram-negative strain *Helicobacter pylori* (ATCC 43504) and yeast strain *Candida albicans* (ATCC 10231) were used to evaluate the antimicrobial potency. The microbial strains and cell line (A549) were provided by the Microbiology and Immunology Department and the Cell Culture Unit, respectively, at the Faculty of Pharmacy, October 6 University, Giza, Egypt.

### *2.2. Methods*

#### 2.2.1. Plant Extraction

Frankincense extract (Fr-extract) was prepared from gum oleoresin (1 Kg) with ethanol (70%) via Soxhlet extraction. The solvent was filtered and evaporated (40–50 °C) until dryness (Rotavapor® R-300, BÜCHI, Switzerland) [29]. Fr-extract was subjected to HPLC quantification, SNEDS formulation, and in vitro biological studies.

#### 2.2.2. HPLC Standardization

Based on pentacyclic boswellic acids, HPLC standardization of BAs was performed on Shimadzu (LC-20AT series) equipped with multiple wavelength detector (SPD-20 AD) and prontosil ODS $C_{18}$ column (5 μm, 15 cm × 4.5 mm). Quantification was performed using an isocratic elution with acetonitrile:water:acetic acid (99:1:0.01 v/v/v) with 1 mL min$^{-1}$ flow rate [2]. Ultraviolet monitoring was carried out at 210 and 260 nm, and the retention times ($R_t$) were determined [30]. The validation method was conducted according to ICH guidelines [31].

### *2.3. Formulation Development and Optimization of Fr-SNEDS*

#### 2.3.1. Screening of Different Excipients for Development of Fr-SNEDS

A solubility study was performed to select the most suitable excipients for SNEDS formulation. The solubility of the standardized Fr-extract in various oils (Labrafil® M 1944, Capryol 90™, Myritol 318, and IPM), in 10% solutions of different surfactants (Labrasol®, Gelucire® 44/14, Kolliphor® RH40, and Tween 80), and in different cosurfactants (PG, PEG400, and ethanol) was investigated. An accurate weight of Fr-extract (0.2 g) was mixed with each excipient (1 mL) and stirred using a vortex mixer (10 min), and the mixtures were shaken in screw-cap glass vials (24 h, 25 ± 2 °C). The formed mixtures were visually inspected and classified as completely soluble, slightly soluble, or insoluble. The suitable oil was selected based on the solubility study. To select the appropriate surfactant and cosurfactant, besides the solubility study results, the ability of the surfactant and the cosurfactant to emulsify the selected oil was screened. Briefly, oil and surfactant were mixed (1:1 w/w), then water was added dropwise with stirring until the first turbidity. The maximum amount of water incorporated into the mixture served as the standard in the final selection of the surfactant. The cosurfactants were similarly screened with the selected surfactant [32]. Water incorporation % was calculated by the following equation:

Water incorporation (%)

$$= [(\text{weight of water added})/(\text{weight of mixture} + \text{weight of water added})] \times 100$$

### 2.3.2. Experimental Design for Development and Optimization of Fr-SNEDS

The used oil, surfactant, and cosurfactant were selected based on the screening study. DoE was conducted for the development and statistical optimization of Fr-SNEDS formulation. A mixture simplex lattice design was employed using the Design-Expert software (Version 11.1.2.0; Stat-Ease Inc., USA). The independent variables were represented by the mixture's three components (oil (A), surfactant (B), and cosurfactant (C)). The low and high values for each component were designated as (0.05 to 0.9). According to the proceeded simplex design, 13 formulations were prepared. Figure 1 shows the schematic presentation of the experimental domain in the ternary diagram, and the conducted experimental points. Droplet size (DS) (Y1) and polydispersity index (PDI) (Y2) were set as dependent variables. Analysis of variance (ANOVA) was carried out to estimate the significance ($p < 0.05$) of each factor. The best fitting mathematical model was selected based on the comparison of several statistical parameters, namely the multiple correlation coefficient ($R^2$) and adjusted $R^2$. The factor effect plots and response surface plots were generated using the same software. For each formulation, an appropriate amount of the Fr-extract to prepare Fr-SNEDS (2%) was dissolved in the specified composition of the oil, surfactant, and cosurfactant by stirring (15 min) until completely dissolved and then stored in closed containers before further evaluation. Each formulation was diluted 100 times by mixing with double distilled water and the mean DS and PDI of the formed dispersion were determined by dynamic light scattering (DLS) technique using a Zetasizer (ZS, Malvern, UK) at (25 ± 2°C). A laser beam (at 632 nm) was applied, and the light scattering was monitored at a fixed angle of 173° and the run time was selected to be automatic to adopt best reading of the sample. The DS was expressed as Z-average (d. nm), which represents the grand average of all the intensities that the DLS picks up. The optimized formulation was determined by the software using the desirability function to attain minimum DS and PDI.

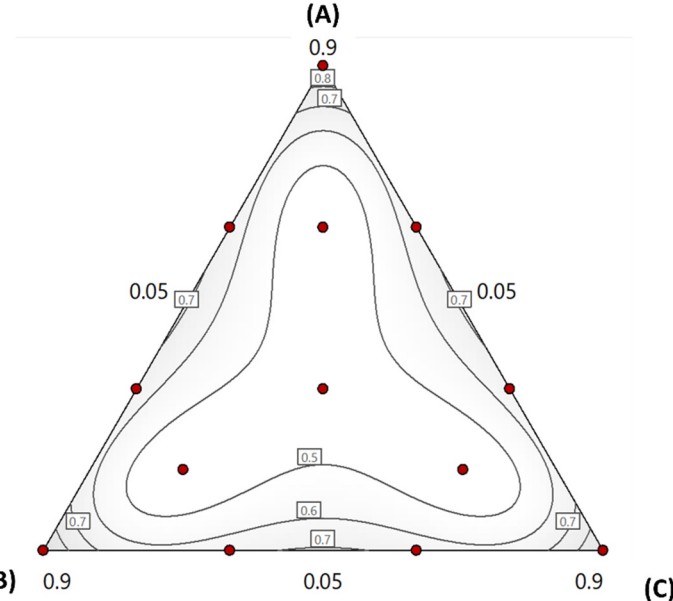

**Figure 1.** Simplex lattice design used for optimization of Fr-SNEDS (oil (**A**), surfactant (**B**), and cosurfactant (**C**)).

### 2.4. Characterization of the Optimized Fr-SNEDS

### 2.4.1. Visual inspection, DS, PDI, and Zeta potential (ZP) measurements

The optimized Fr-SNEDS was inspected visually against strong light for optical transparency and homogeneity. DS and PDI were determined as previously mentioned, and ZP was measured. Measurements were performed in triplicates at 25 °C.

2.4.2. Assessment of Self-Emulsification Efficiency

The rate of emulsification is an important index for the assessment of the efficacy of SNEDS. The test was performed using a USP II dissolution apparatus (Distek 2500, North Brunswick Township, NJ USA). A watch glass containing formulation (0.5 g) was introduced into HCl (0.1 N, pH 1.2, 500 mL) at $37 \pm 0.5$ °C with gentle agitation (100 rpm). The performance of self-emulsification was assessed visually by monitoring the time required for the complete disappearance of the formulation and formation of a clear dispersion [33,34]. The formed dispersion was evaluated by measuring the percent transmittance using a UV-vis spectrophotometer at 638 nm (UV-160A, Shimadzu, Japan) against double distilled water as the blank [35]. All determinations were performed in triplicates.

2.4.3. Thermodynamic Stability

The optimized Fr-SNEDS was subjected to 2 stress tests to assess its stability and ability to withstand thermal changes. The optimized Fr-SNEDS was subjected to centrifugation test (3000 rpm, 30 min) and freezing–thawing cycles (3 complete cycles, each of 24 h at −5 °C, followed by 24 h at 25 °C). At the end of each test, the formulation was examined visually for phase separation or extract precipitation.

2.4.4. Study of Surface Morphology by Transmission Electron Microscopy (TEM)

The morphology of the optimized Fr-SNEDS was investigated by 200 kV Field Emission (FE) TEM (JEM-2100, JEOL Ltd., Tokyo, Japan) integrated in PC control system enables scanning images of a sample at nanometer resolution. A drop of a diluted aqueous dispersion of the formulation was dropped onto copper-coated carbon grids, dried, then stained with phosphotungstic acid (2% w/v).

*2.5. Assessment of Antimicrobial Activity*

2.5.1. Preparation of the Inoculum

Microbial biofilm strains *B. subtilis*, *M. luteus*, *S. aureus*, *S. epidermidis*, and MRSA were sub-cultured aerobically overnight (35 °C) in Mueller–Hilton agar. *C. albicans* and *H. pylori* were inoculated in sabouraud agar (48 h, 25 °C), and microaerophilic (5% $CO_2$, 72 h, 37 °C) in supplemented Columbia blood agar. The microbial growth was diluted to attain 0.5 MacFarland turbidity standard using a spectrophotometer (590 nm).

2.5.2. Sensitivity Test

The antimicrobial susceptibility of the microbial biofilm strains were determined against antimicrobial agents (positive control), Fr-extract, and Fr-SNEDS using Kirby Bauer disk diffusion [36,37]. Fr-extract (10 mg mL$^{-1}$ in DMSO), Fr-SNEDS dispersion in sterile water (equivalent to 10 mg mL$^{-1}$ Fr-extract), and plain SNEDS dispersion were sterilized through a millipore filter (0.22 mm), then loaded over sterile discs (6 mm) on supplemented petri dishes with respective strains in a concentration of 0.5 MacFarland standard. The plates were incubated as respective conditions and the zones of inhibition (mm) were measured in triplicates.

2.5.3. Minimum Inhibitory Concentrations (MICs) and Minimum Biocidal Concentrations (MBC)

The lowest concentration that inhibits the microbial growth (MIC) and the lowest concentration required to kill a microbe over a fixed period (18 or 24 h) (MBC) for Fr-extract and Fr-SNEDS were determined using broth microdilution method [37]. Separately, serial dilutions of samples (200 to 5 μg mL$^{-1}$) were inoculated with respective microbes (10 μL), evaluated, and microbial counts (CFU mL$^{-1}$) were determined in triplicates using an ELISA plate reader (Infinite F50 Tecan-Sweden, 620 nm) [38].

*2.6. In Vitro Antiproliferative Assay*

Antiproliferative assay (as $IC_{50}$) of the standardized Fr-extract and Fr-SNEDS were performed using the MTT assay [39]. Briefly, A549 cells ($1 \times 10^3$ cells mL$^{-1}$) were plated in 96-well plates and allowed to attach for 24 h, resulting in a log phase growth at the time of drug treatment. Serial dilutions (100–1.56 μg mL$^{-1}$) of the Fr-extract in DMSO, Fr-SNEDS, and plain SNEDS dispersions in distilled water were added to the cells. Cells were also treated with each component of the plain SNEDS formulation individually in the same concentration range used in the formulation. After treatment, MTT (10 μL) was added to each well, incubated (4 h, at 37 °C), and the produced formazan was solubilized in DMSO (100 μL). The absorbance was measured using a microplate ELISA reader (FLU-Ostar Omega, BMG, Labtech, Germany) at 490 nm. The percentage of relative cell viability was calculated using the following equation: Cell viability % = [Absorbance of treated cells/Absorbance of control cells)] × 100.

Each concentration was performed in triplicates and the data were analyzed using GraphPad Prism 6 (La Jolla, CA, USA). The values of cell viability % versus a series of sample concentrations were then plotted using non-linear regression analysis of Sigmoidal dose–response curve generated with the equation Y = B + (T − B)/1 + 10((LogEC50-X) × Hill Slope), where Y = percent cell, B = minimum percent cell, T = maximum percent cell, X = logarithm of compound, and Hill Slope = slope factor or Hill coefficient [29].

*2.7. Stability Study*

The optimized Fr-SNEDS was stored in a dark place for 12 months at 25 ± 2 °C and 65% relative humidity. The effect of storage on clarity, DS, PDI, ZP, surface morphology, and self-emulsification efficiency was evaluated.

## 3. Results

*3.1. Standardization of BAs in fr-Extract*

The quantification of bioactive BAs is essential for their use in dietary supplements. BAs, a mixture comprised of four major triterpenes, namely $\beta$-boswellic acid ($\beta$-BA), 3-acetyl-$\beta$-BA, 11-keto-boswellic acid (KBA), and 3-acetyl-11-keto-boswellic acid (AKBA),and two minors, namely $\alpha$-boswellic acid ($\alpha$-BA), and 3-acetyl-$\alpha$-BA, were isolated from the oleogum resin of *Boswellia* species [1]. KBA and AKBA are considered the most active BAs. It is noteworthy that the $R_t$ of KBA and $\beta$-BA in the performed HPLC analysis are 2.5 and 6.5 min, respectively (Table 1), in agreement with those reported (2.6–3.4 and 7.5–8.6 min, respectively) [2]. However, their acetyl derivatives $R_t$ were 3.7 and 7.9 min, respectively (Figure 2), also in agreement with those reported (3.5–4.2 and 11.5–12.2 min, respectively) [2].

**Table 1.** HPLC determination of BAs in frankincense extract.

| Compound | $R_t$ (min) | Content % |
|---|---|---|
| 11-keto-boswellic acid | 2.5 | 16.25% (major) |
| Acetyl-11-keto-boswellic acid | 3.7 | 11.8% (major) |
| $\alpha$-boswellic acid | 5.8 | 2.19% (minor) |
| $\beta$-boswellic acid | 6.5 | 8.76% (major) |
| Acetyl-$\alpha$-boswellic acid | 7.5 | 2.5% (minor) |
| Acetyl-$\beta$-boswellic acid | 7.9 | 3.46% (minor) |
| | Total content | 44.96% |

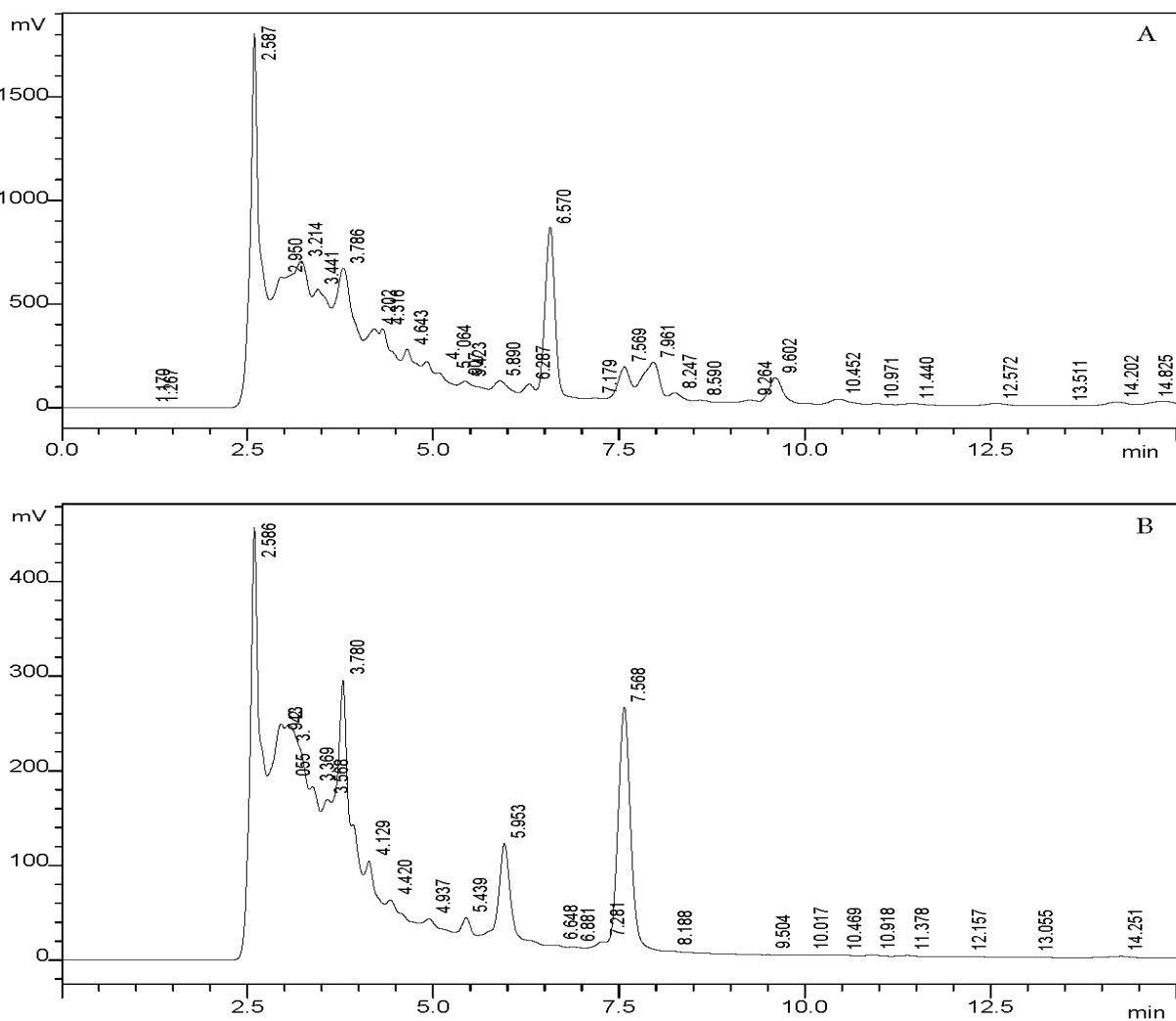

**Figure 2.** Analytical HPLC-UV chromatograms of frankincense extract at (**A**) 210 nm and (**B**) 260 nm.

### 3.2. Formulation Development and Optimization of Fr-SNEDS

### 3.2.1. Screening of Excipients

We investigated the ability of the excipients to dissolve a known fixed amount of the Fr-extract (10 times the loaded drug amount) in the tested excipients. Among the tested oils, only Capryol™ 90 showed a good Fr-extract solubility (Table 2). After the selection of the oil phase, the goal was to identify the surfactant and cosurfactant that have both good Fr-extract solubility and emulsification ability of Capryol™ 90. Gelucire® 44/14, PG, PEG400, and ethanol showed good Fr-extract solubility (Table 2). However, the main selection perspective for the surfactant and cosurfactant was the emulsification ability of the selected oil rather than Fr-extract solubility. Gelucire® 44/14 had a superior ability to emulsify Capryol™ 90, followed by Tween 80 and Labrasol®, whereas Kolliphor® RH40 showed the lowest emulsification ability for the oil based on the water incorporation percent (Figure 3). The selection of cosurfactant was also based on emulsification ability, where ethanol showed the highest water incorporation % when combined with Gelucire® 44/14 as a surfactant. Finally, Capryol™ 90 was selected as the oil phase because it showed a good Fr-extract solubility, and Gelucire® 44/14 and ethanol were selected as surfactant and cosurfactant as they recorded good Fr-extract solubility and good Capryol™ 90 emulsification ability.

**Table 2.** Solubility study observations of frankincense extract in different oils, surfactants (10% solutions), and cosurfactants.#

| Excipient | Observed Solubility |
|---|---|
| Labrafil® M 1944 CS | Slightly soluble |
| Capryol™ 90 | Completely soluble |
| IPM | Insoluble |
| Myritol 318 | Insoluble |
| Tween 80 | Slightly soluble |
| Gelucire® 44/14 | Completely soluble |
| Labrasol® | Slightly soluble |
| Kolliphor® RH40 | Insoluble |
| PG | Completely soluble |
| PEG400 | Completely soluble |
| Ethanol | Completely soluble |

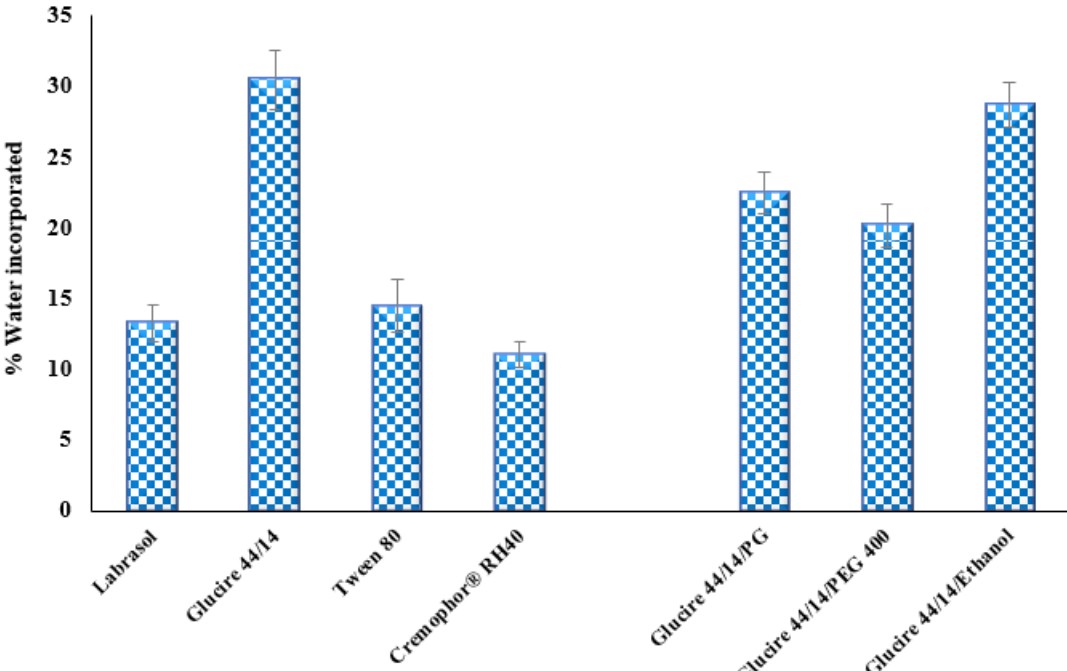

**Figure 3.** Percentage of water incorporated into mixtures of Capryol™ 90 and different surfactants and surfactant/cosurfactant mixtures (mean ± SD, n = 3).

### 3.2.2. Experimental Design

Based on the screening study, Capryol™ 90, Gelucire® 44/14, and ethanol were used for Fr-SNEDS formulation. A simplex lattice design was used to investigate the effect of the independent variables (oil ($X_1$), surfactant ($X_2$), and cosurfactant ($X_3$) concentrations) on the dependent variables (DS (Y1) and PDI (Y2)). The experimental matrix for the prepared 13 Fr-SNEDS formulations and the corresponding responses of DS and PDI is presented in Table 3.

**Table 3.** Experimental runs suggested by simplex lattice mixture design approach for the optimization of Fr-SNEDS and the determined responses. Data are expressed as mean ± SD (*n* = 3).

| Formulation No. | Factors | | | Responses | |
|---|---|---|---|---|---|
| | Oil (A) Capryol™ 90 | Surfactant (B) Gelucire® 44/14 | CoS (C) Ethanol | DS (nm) | PDI |
| 1 | 0.19 | 0.19 | 0.60 | 237 ± 6 | 0.359 ± 0.01 |
| 2 | 0.60 | 0.19 | 0.19 | 456 ± 25 | 0.67 ± 0.04 |
| 3 | 0.049 | 0.88 | 0.049 | 21 ± 0.4 | 0.311 ± 0.02 |
| 4 | 0.33 | 0.60 | 0.049 | 296 ± 6 | 0.478 ± 0.02 |
| 5 | 0.049 | 0.60 | 0.33 | 15.5 ± 0.17 | 0.249 ± 0.015 |
| 6 | 0.60 | 0.33 | 0.049 | 470 ± 15 | 0.677 ± 0.07 |
| 7 | 0.33 | 0.049 | 0.60 | 372 ± 21 | 0.562 ± 0.06 |
| 8 | 0.88 | 0.049 | 0.049 | 536 ± 28 | 0.672 ± 0.06 |
| 9 | 0.60 | 0.049 | 0.33 | 460 ± 35 | 0.736 ± 0.07 |
| 10 | 0.049 | 0.33 | 0.60 | 47 ± 0.5 | 0.30 ± 0.01 |
| 11 | 0.049 | 0.049 | 0.88 | 193 ± 2 | 0.258 ± 0.01 |
| 12 | 0.19 | 0.60 | 0.19 | 99 ± 11 | 0.40 ± 0.07 |
| 13 | 0.33 | 0.33 | 0.33 | 314 ± 5 | 0.59 ± 0.02 |

* Each formulation was loaded with 2% extract (0.02 g to obtain 1 g Fr-SNEDS).

Model Statistical Analysis and ANOVA

The statistical analysis of simplex lattice design was performed by multiple linear regression analysis using the software. After statistical analysis for various models to determine the best fit model, DS and PDI were individually fitted to the quadratic model. Model signification and suitability were tested by analysis of variance (ANOVA) and the model statistics are illustrated in Table 4. ANOVA analysis showed significant *p*-values (<0.0001) for both DS and PDI, and F-values of 127.14 and 39.64 for DS and PDI, respectively, that implied the model significance. For each response, the model was validated by the reasonable agreement of the predicted and the adjusted $R^2$ values (i.e., the difference is less than 0.2). The actual model $R^2$ and predicted and adjusted $R^2$ values of the DS($Y_1$) were 0.9891, 0.9813, and 0.9684, respectively. Considering PDI ($Y_2$), the actual model $R^2$ and predicted and adjusted $R^2$ values were of the PDI were 0.9659, 0.9415, and 0.8370, respectively. Moreover, the adequate precision was higher than 4 (31.37 and 15.94 for DS and PDI, respectively), indicating an acceptable signal to noise ratio and consequently, the ability of the model to navigate the design space [40].

**Table 4.** ANOVA for dependent variables from simplex lattice design.

| | Source | Sum of Squares | df | Mean Square | F-Value | *p*-Value |
|---|---|---|---|---|---|---|
| DS | Model | $4.031 \times 10^5$ | 5 | 80620.85 | 127.14 | <0.0001 * |
| | Linear Mixture | $3.763 \times 10^5$ | 2 | $1.881 \times 10^5$ | 296.68 | <0.0001 * |
| | $X_1 X_2$ | 13167.92 | 1 | 13167.92 | 20.77 | 0.0026 * |
| | $X_1 X_3$ | 3790.31 | 1 | 3790.31 | 5.98 | 0.0444 * |
| | $X_2 X_3$ | 7353.97 | 1 | 7353.97 | 11.60 | 0.0114 * |
| | $R^2$ = 0.9891, | Prediction $R^2$ = 0.9813, | | Adjusted $R^2$ = 0.9684, | Adeq. Precision = 31.3709 | |
| PDI | Model | 0.3678 | 5 | 0.0736 | 39.64 | <0.0001 * |
| | Linear Mixture | 0.3238 | 2 | 0.1619 | 87.26 | <0.0001 * |
| | $X_1 X_2$ | 0.0101 | 1 | 0.0101 | 5.42 | 0.0528 |
| | $X_1 X_3$ | 0.0367 | 1 | 0.0367 | 19.79 | 0.0030 * |
| | $X_2 X_3$ | $8.837 \times 10^{-6}$ | 1 | $8.837 \times 10^{-6}$ | 0.0048 | 0.9469 |
| | $R^2$ = 0.9659, | Prediction $R^2$ = 0.9415, | | Adjusted $R^2$ = 0.8370, | Adeq. Precision = 15.944 | |

* Significant (*p* < 0.05).

The relation between dependent and independent variables was established using mathematical relationships. To study the effects, regression equations were generated for each response; the values of the coefficients of the independent variables (oil ($X_1$), surfactant ($X_2$), and cosurfactant ($X_3$)) were reflecting the effects of these variables on each response. A positive coefficient represents a synergistic effect, whereas a negative coefficient indicates an opposite effect on the response; moreover, a greater coefficient indicates that the independent variable has a stronger effect on the response. Coefficients with more than one term indicate interaction; when two factors change simultaneously, a factor can produce a different degree of response [41].

Effect of Formulation Composition on DS

DS values of the prepared Fr-SNEDS formulations were found to be in the range of 15.5 to 536 nm (Table 3). The polynomial equation obtained for DS was:

$$DS = + 535.50\, X_1 - 36.29\, X_2 + 197.038\, X_3 + 644.66\, X_1 X_2 + 345.86\, X_1 X_3 - 481.763\, X_2 X_3$$

ANOVA analysis report for DS indicated that ($X_1$), ($X_2$), ($X_1 X_2$), ($X_1 X_3$), and ($X_2 X_3$) are significant model terms ($p < 0.05$). As recorded in Table 4, the oil ($X_1$) and cosurfactant ($X_3$) showed a positive effect on DS while surfactant ($X_2$) had a negative effect. The interaction effects of ($X_1 X_2$) and ($X_1 X_3$) were positive, while the interaction effect of ($X_2 X_3$) was negative (Figure 4A–C).

The effect of concentrations of oil, surfactant, and cosurfactant (independent variables) on DS is illustrated in Figures 5A and 5B, showing two- and three-dimensional contour and response surface plots. It was observed that the highest DS values are presented at the top of the figure (orange area) at increased Capryol™ 90 percentage, while the lowest values (blue area) are found as the Gelucire® 44/14 percentage increased. When the concentration of Gelucire® 44/14/ethanol mixture exceeded 75% of the system, DS values less than 200 nm are presented, indicating good emulsification efficiency.

Effect of Formulation Composition on PDI

The polydispersity index is an important parameter to evaluate the diameter distribution; lower PDI values indicate uniformity of droplet size within the formulation and assure its stability [42]. PDI values of the prepared Fr-SNEDS were found to be in the range of 0.249 to 0.736. The obtained polynomial equation for PDI was:

$$PDI = + 0.6605 X_1 + 0.2451\, X_2 + 0.171 X_3 + 0.563 X_1 X_2 + 1.076 X_1 X_3 - 0.0167 X_2 X_3$$

ANOVA analysis report (Table 4) indicated that ($X_1$), ($X_2$), ($X_3$), and ($X_1 X_3$) are significant model terms ($p < 0.05$). Considering the above equation of PDI, the three factors ($X_1$, $X_2$, and $X_3$) had a positive effect—the oil showed the highest positive value, followed by the surfactant, and the least value was recorded for the cosurfactant. The interaction effects of ($X_1 X_2$) and ($X_1 X_3$) were positive, while the interaction effect of ($X_2 X_3$) was negative (Figure 4D,E,F).

The effect of oil, surfactant, and cosurfactant on PDI is illustrated in Figure 5C,D, using two- and three-dimensional contour and response surface plots. PDI was significantly decreased by decreasing Capryol™ 90 percentage, whereas Gelucire® 44/14 and ethanol percentages were of low significance. An optimal range of the oil percentage for achieving a minimum PDI was around 5%, with different percentages of surfactant and cosurfactant.

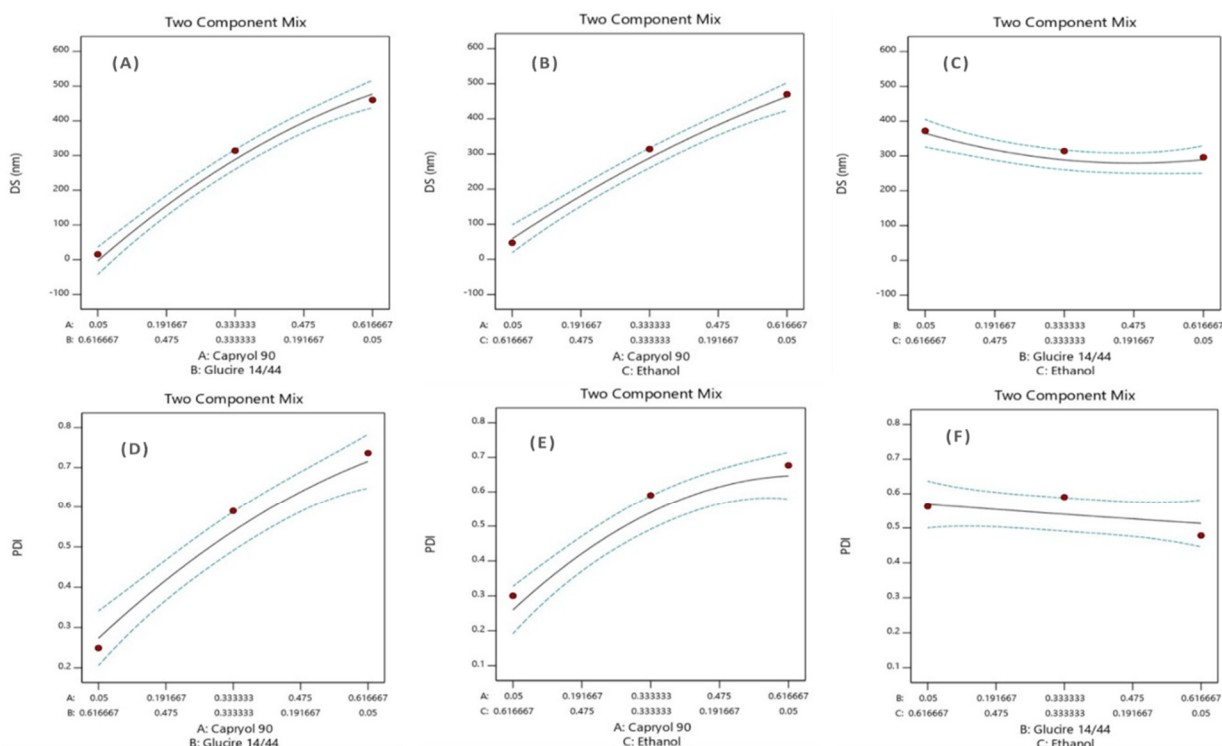

**Figure 4.** Factor effect plots showing the effect of two-component mix (Capryol™ 90/Gelucire® 44/14, Capryol™ 90/Ethanol, and Gelucire® 44/14/Ethanol): (**A-C**) on droplet size (DS) and (**D-F**) on poly dispersity index (PDI).

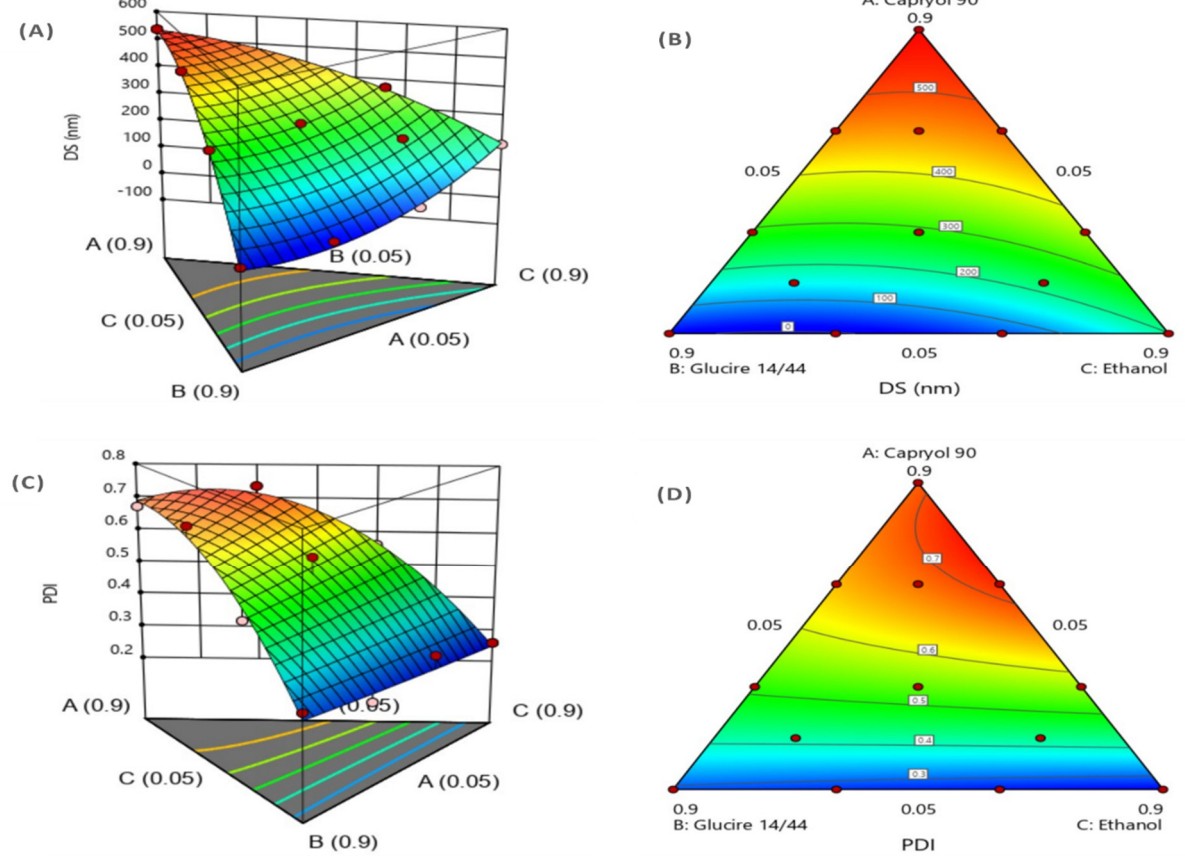

**Figure 5.** Response surface plots for the effect of Fr-SNEDS compositions (Capryol™ 90, Gelucire® 44/14, and Ethanol): (**A,B**) on droplet size (DS) and (**C,D**) on poly dispersity index (PDI).

Formulation Optimization

The optimization process was performed using the desirability function to optimize DS and PDI to be minimized simultaneously. After applying the constraints cited in Table 5, the composition of the optimized Fr-SNEDS formulation was obtained with a desirability value of 0.964, indicating that all goals are met perfectly. The composition of the optimized formulation was 5%, 48.7%, and 46.3% for Capryol™ 90, Gelucire® 44/14, and ethanol, respectively. The optimized formulation was prepared and evaluated for DS and PDI. The similarity between the predicted values (15.5 and 0.267 for DS and PDI, respectively) and the observed values (17.92 and 0.297 for DS and PDI, respectively) confirmed the validity of the generated mathematical equations and the used model.

**Table 5.** Independent variables and respective limits for Fr-SNEDS formulations, model summary statistics of a quadratic model, constraints for optimization, factors levels for optimized Fr-SNEDS formulation, and their predicted and observed values.

| Factors (Independent Variables) | Design Constraints | | |
|---|---|---|---|
| | Low Limit (+0) | Upper Limit (+1) | Optimized Level |
| A: Capryol™ 90 | 0.05 | 0.9 | 0.050 |
| B: Gelucire® 44/14 | 0.05 | 0.9 | 0.487 |
| C: Ethanol | 0.05 | 0.9 | 0.463 |
| Responses (Dependent Variables) | Constraints | 95% Prediction Interval (Low/High) | Predicted | Observed |
| DS (nm) | Minimize | −39.08 /70.08 | 15.500 | 17.92 ± 0.2 |
| PDI | Minimize | 0.173/0.36 | 0.267 | 0.297 ± 0.015 |

*3.3. Characterization of the Optimized Fr-SNEDS:*

3.3.1. Visual Inspection, DS, PDI, and ZP Measurements and Assessment of Self-Emulsification

Visual inspection of the optimized Fr-SNEDS showed clear, transparent, and homogeneous fluid with no signs of phase separation or precipitation. Fr-SNEDS showed a small size (17.9 ± 0.2 nm) and a small value of PDI (0.3 ± 0.015), indicating a monodispersed population and a high uniformity among particles. Zp value was −14.5 ± 0.236 mV. The absolute Zp values >8–9 mV are essential for the system's stability. The emulsification time for the optimized Fr-SNEDS was less than one min and the % transmittance of the formed dispersion was 97.8 ± 0.2%, which reflected good self-emulsification of the system.

3.3.2. Thermodynamic Stability

The optimized Fr-SNEDS showed good physical stability, as no changes in physical appearance, phase separation, or precipitation of the extract were observed.

3.3.3. Surface Morphology by TEM

The morphology of the formed dispersion globules of the optimized Fr-SNEDS is illustrated in the photograph of TEM (Figure 6A). The globules had a uniform spherical shape and a homogeneous size distribution. The diameters of the globules were in a range of 16–29 nm that was comparable with the DLS size results.

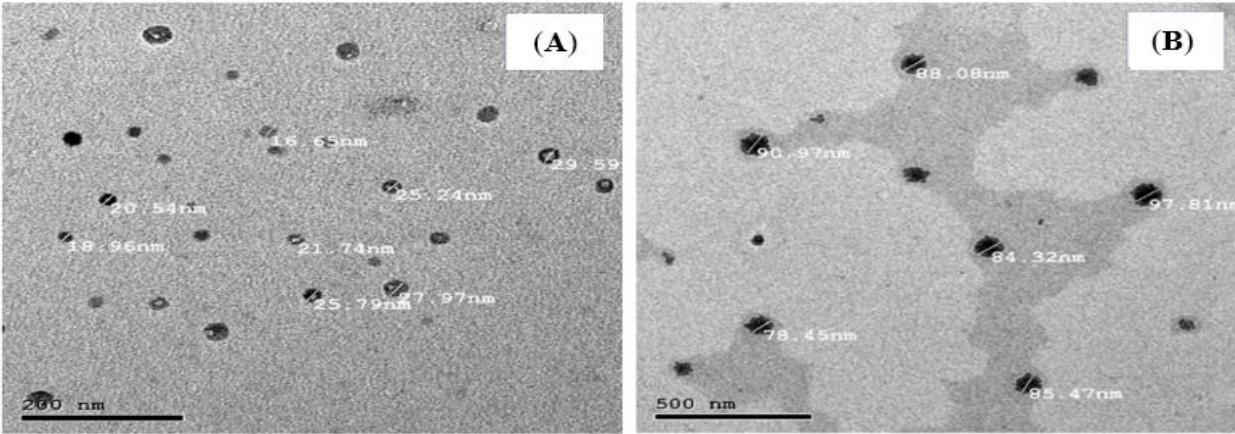

**Figure 6.** TEM photographs of Fr-SNEDS freshly prepared (**A**) and stored (**B**) formulation under ambient conditions for 12 months.

### 3.4. Assessment of Antimicrobial Activity

The comparative assessment of the antimicrobial activity of Fr-extract and Fr-SNEDS revealed that both were potentially effective in suppressing microbial growth of all tested organisms with variable potencies (Table 6). The highest zone of inhibition was recorded against *H. pylori* and the lowest was against *M. luteus*. Fr-SNEDS exhibited higher zones of inhibition than the antimicrobial agents. Depending on the inhibition zone diameters, Fr-SNEDS showed a strong activity against all tested organisms (inhibition zone > 20 mm), except *M. luteus* and *C. albicans* as it showed moderate (12 mm < inhibition zone < 20 mm) activities [43], and no explanation is suggested. More expressive data on the antimicrobial properties were obtained through the determination of MIC and MBC values of Fr-extract and Fr-SNEDS (Figure 7). Fr-SNEDS exhibits lower MIC and MBC values compared with Fr-extract, where *S. aureus* and *H. pylori* were the most susceptible bacteria (MIC values of 5 and 25 $\mu$g mL$^{-1}$, and MBC values of 5 and 100 $\mu$g mL$^{-1}$ respectively), while *M. luteus* was the most resistant strain to Fr-SNEDS (MIC = 175 $\mu$g mL$^{-1}$). In conclusion, the results showed a strong activity of Fr-SNEDS against *S. aureus*, *S. epidermidis*, *B. subtilis*, and a good activity against MRSA, which are conjugated organisms with lung cancer. In addition, there is a growing body of evidence supports an association between *H. pylori* infection with lung cancer [44,45].

**Table 6.** Zones of inhibition of Fr-extract, Fr-SNEDS, and plain SNEDS formulation against different bacterial strains compared with antimicrobial agents. Data are expressed as mean ± SD (*n* = 3).

| Microbial Species | Fr-Extract | Fr-SNEDS | Plain SNEDS | Antimicrobial Agents * | | | |
|---|---|---|---|---|---|---|---|
| | | | | F | AMC | CFPM | VA |
| *S. aureus* | 19 ± 1.1 | 23 ± 1.3 | 9 ± 0.3 | 20 ± 1.3 | 34 ± 2.1 | 25 ± 1.9 | 16 ± 0.7 |
| *S. epidermidis* | 19 ± 1.5 | 24 ± 1.5 | 9 ± 0.4 | 19 ± 0.9 | NA | 20 ± 1.7 | 14 ± 0.6 |
| *B. subtilis* | 22 ± 0.2 | 24 ± 0.7 | 9 ± 0.3 | 25 ± 1.9 | NA | 20 ± 1.4 | 17 ± 0.5 |
| *M. luteus* | 20 ± 0.5 | 12 ± 0.3 | NA | 14 ± 0.8 | 18 ± 0.3 | 23 ± 1.3 | 15 ± 0.8 |
| *MRSA* | 22 ± 1.8 | 22 ± 1.4 | 9 ± 0.3 | ND | ND | R | 17 ± 0.9 |
| *H. pylori* ** | 25 ± 0.2 | 28 ± 1.1 | 10 ± 0.6 | ND | ND | ND | NA |
| *C. albicans* *** | 22 ± 1.1 | 14 ± 1.0 | NA | NA | NA | NA | NA |

DMSO, Capryol™ 90 and Gelucire® 44/14 showed no zones of inhibitions. NA: Not active, ND: Not determined, R: Resistant. * Antimicrobial agents: F (Nitrofurantoin-300 $\mu$g/disc), AMC (Amoxicillin/Clavulanic acid 20/10 $\mu$g/disc), CFPM (Cefepime-30 $\mu$g/disc), VA (vancomycin 30 $\mu$g/disc). Fluconazole (150 mg) and Metronidazole (5 $\mu$g/disc) were induced zone of inhibition of 19 and 27 mm against ** and ***, respectively.

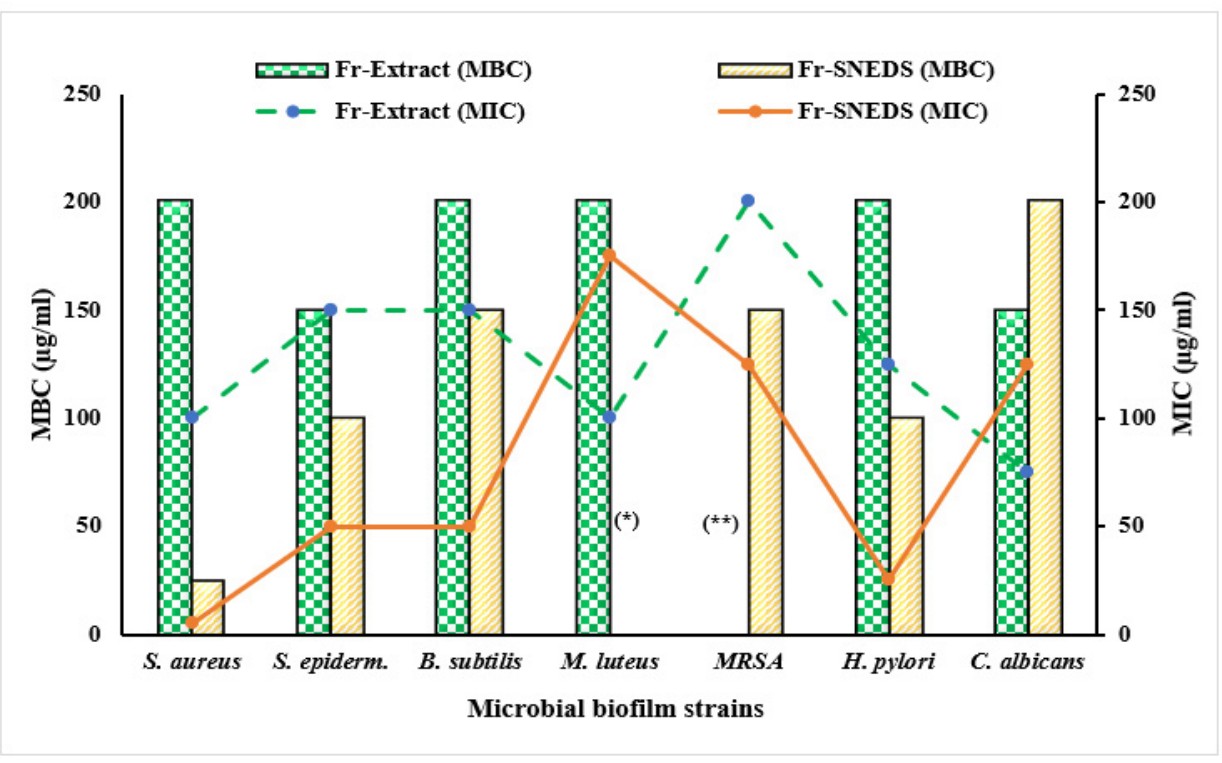

**Figure 7.** MICs and MBCs (µg mL$^{-1}$) of Fr-extract and Fr-SNEDS formulation against different microbial strains. (*) Fr-SNEDS MBC against *M. luteus* was not determined, (**) Fr-extract MBC against *MRSA* was >200.

### 3.5. Antiproliferative Study

The antiproliferative testing was performed on lung cell line (A549) based on the antimicrobial findings, which showed good activity against the bacteria associated with lung cancer. Fr-SNEDS showed a significant antiproliferative effect when compared with Fr-extract (IC$_{50}$: 20.49 and 109.5 µg mL$^{-1}$, respectively). It is obvious that the nano-dispersion enhanced the antiproliferative effect of Fr-extract. Additionally, the plain formulation exerted antiproliferative effects (IC$_{50}$: 78.61 µg mL$^{-1}$) when used in concentrations above 50 µg mL$^{-1}$. The cell viability was evaluated individually for each component in its respective concentration used in the formulation (Capryol™ 90, Gelucire® 44/14, and ethanol) and no significant cell death was observed.

### 3.6. Stability Study

The results of the stability study suggested a good physical stability of the optimized Fr-SNEDS. The formulation almost retained its properties—the stored formulation had DS close to the fresh one, PDI and ZP were slightly increased (Table 7). The self-emulsification parameters proved that the system dispersed within less than one min and the obtained dispersion was clear, with % transmittance of 97.8%. TEM photograph of the stored Fr-SNEDS showed homogeneously distributed spherical nano-sized globules (Figure 6B).

**Table 7.** The physicochemical parameters for fresh and stored Fr-SNEDS. Data are expressed as mean ± SD (*n* = 3).

| Fr-SNEDS | DS (nm) | PDI | ZP (mV) | Assessment of Self-Emulsification | |
| --- | --- | --- | --- | --- | --- |
| | | | | Emulsification Time | % Transmittance |
| Freshly prepared | 17.9 ± 0.2 | 0.3 ± 0.015 | -14.5 ± 0.236 | <1 min | 97.8 ± 0.2 |
| Stored for one year | 22.3 ± 0.7 | 0.5 ± 0.018 | -22.4 ± 0.781 | <1 min | 98.1 ± 0.1 |

## 4. Discussion

Despite the various therapeutic potentials of phytoconstituents, there are some limitations in their therapeutic applications due to low solubility, poor permeability, and difficulty of their formulation into suitable dosage forms. The use of nanotechnology in formulation could solve most of these encountered issues. In this work, we developed SNEDS formulation of a standardized Fr-extract; the physicochemical characters and stability of the optimized formulation were investigated. In addition, we compared the performance of Fr-SNEDS to the unformulated Fr-extract through in vitro screening of the antimicrobial and antiproliferative activities to investigate the influence of nano formulation and elucidate the possible potential therapeutic applications of this formulation.

Frankincense is a natural resin with different therapeutic properties, including antimicrobial and cytotoxic activities, mainly due to its BA content. In this study, we determine the BA content in Fr-extract using HPLC. The HPLC analysis of the Fr-extract shows 44.96% total content of Bas, where KBA has the major content, followed by AKBA and *β*-BA (16.25%, 11.8%, and 8.76%, respectively). Our results agree with those reported, including that AKBA content of *Boswellia* species ranged from 0.05–5.40% [46]. In another report, AKBA content was estimated as 7.35% of gum resin [30,47]. Our findings indicated that the total content of the BAs is within 50%, in agreement with previous works [1,48].

Formulation development was a challenge due to the physical nature of the Fr-extract as it was a very sticky semisolid mass and insoluble in water. Excipients screening for development of Fr-SNEDS was based on the solubility study as loading of the poorly soluble Fr-extract will depend mainly on its solubility in different formulation components. Selection of the oily phase was based on Fr-extract solubility, while selection of surfactants and cosurfactants was based on both Fr-extract solubility and emulsification ability for the selected oil. Capryol™ 90, Gelucire® 44/14 and ethanol were selected as oily phase, surfactant, and cosurfactant, respectively. Although the HLB values of the investigated surfactants were in the range of 14 to 16, there was a difference in their emulsification ability for Capryol™ 90. It was reported that emulsification is influenced by HLB value, molecular structure, and chain length of the surfactant [17,25]. The cosurfactant further lowers the interfacial tension, improving the stability of the formed nanoemulsion dispersion. A simplex lattice design was used to investigate the effect of formulation variables on DS and PDI. Statistical analysis revealed that oil concentration had the most pronounced positive effect on both DS and PDI, i.e., they increased with increasing oil percentage. In addition, all interactions that involved oil concentration had a positive effect on DS and PDI. Conversely, surfactant concentration imparted a negative effect on DS. This effect has often been explained as being the result of increased surfactant adsorption around the oil–water interface of a droplet and the consequent decreased interfacial tension in the system. Surfactant molecules provide a mechanical barrier to coalescence, resulting in increased spontaneity of the nanoemulsion formation, which facilitates the formation of nanoemulsions with smaller droplets [17,49]. Ethanol as a cosurfactant can decrease interfacial tension between Capryol™ 90 and water, adjust the flexibility of the interfacial membrane, and reduce the required amount of surfactant. The interaction between the surfactant and cosurfactant showed a negative effect on both responses. Consequently, the optimized Fr-SNEDS formulation generated based on the desirability function to obtain minimum DS and PDI was composed of the lowest oil concentration and almost equal surfactant and cosurfactant concentrations. Characterization of the optimized Fr-SNEDS showed good appearance, self-emulsification properties, and stability.

BAs and AKBA inhibit the growth of Gram-positive pathogens. AKBA effectively inhibits the *S. aureus* and *S. epidermidis* biofilms [6]. In our study, the antimicrobial assessment showed that Fr-SNEDS recorded wider zones of inhibition than Fr-extract for all tested organisms except *M. luteus* and *C. albicans*. It is worthy to mention that the plain SNEDS showed small zones of inhibition. The nanosized dispersions can penetrate the bacterial cell wall more efficiently and cause membrane disturbance and cell content disorganization. The smaller the DS is, the greater the extent of uptake through a barrier

membrane, and this could explain the improved antimicrobial efficiency of Fr-SNEDS over Fr-extract [43,50–52]. Fr-SNEDS showed good activity against the bacteria associated with lung cancer. Based on these findings, the antiproliferative testing was performed on lung cell line and the results revealed a superior activity of the optimized Fr-SNEDS over the Fr-extract. Lee et al. [53] reported the enhanced effect of nanoemulsion formulation of tanshinone extract on the inhibition of lung cancer cells (A549); other authors reported the enhanced cytotoxic activity using SNEDS [22,54]. These results revealed that the developed Fr-SNEDS represented the extract in nanosized dispersions, which significantly improved the antimicrobial and antiproliferative activities of Fr-extract.

## 5. Conclusions

Frankincense is widely used in pharmaceutical and nutraceutical preparations. Its bioactivity is based on the content of BAs, where KBA is a major compound. The optimized Fr-SNEDS enhanced the antimicrobial activity against the bacteria associated with lung diseases and improved the antiproliferative activity on human lung cell adenocarcinoma. Our findings suggested that the optimized Fr-SNEDS could be a potential adjuvant therapy for lung cancer. Further in vivo and clinical studies are required to investigate the therapeutic efficiency of such nano-formulations.

**Author Contributions:** S.S.E-M. and O.S.E perform the formulation, formula characterization, revised the manuscript, and designed the project. A.E.E-H. performed the characterization of the extract, revised the manuscript, and conceived and designed the project. W.A.A. perform antimicrobial activity and drafted the manuscript. A.M.S performed the extraction and drafted the manuscript. S.Z.E-E. perform antiproliferative activity and drafted the manuscript. All authors have read and agreed to the published version of the manuscript.

**Funding:** This research received no external funding.

**Institutional Review Board Statement:** Not applicable.

**Informed Consent Statement:** Not applicable.

**Data Availability Statement:** All the data presented during this study are included in the article.

**Conflicts of Interest:** The authors confirm that this article content has no conflict of interest.

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
