# Peer review of "Enhancement of Antimicrobial and Antiproliferative Activities of Standardized Frankincense Extract Using Optimized Self-Nanoemulsifying Delivery System"

_scipharm, doi:10.3390/scipharm89030036_

Round 1

Reviewer 1 Report

The paper entitled "Enhancement of antimicrobial and cytotoxic activities of standardized frankincense extract using optimized self-nanoemulsifying delivery system" is interesting and well written. However, my concerns are related to the in vitro cell experiments, where data is not shown. More, just the simple MTT assay at a one-time point is not sufficient to validate cytotoxicity. Therefore, I recommend the extension of the cell-related experiments before considering this article for publication.

Reviewer 2 Report

Comments and suggestions for authors:

The manuscript prepared by El-Mancy et al. describes the development and optimization of a Boswellia extract formulation. Furthermore, the antimicrobial activity and the in vitro cytotoxicity against a lung cancer cell line was investigated. Before considering for publication the following points should be revised:

Major comments:

  • Chromatographic selectivity: The authors quantified the contents of six boswellic acids by HPLC-UV. According to the chromatograms (Figure 2) the run time was about 15 min. As can be seen clearly from Figure 2 the chromatographic separation is insufficient for selective quantification of individual boswellic acids. The selectivity must be increased by longer run times yielding clearly separated peaks [Paul et al., 2011, Chromatographia; Büchele et al., 2003, J. Chromatogr. B] or using additionally MS (or MS/MS) detection [Katragunta et al., 2019, J. Pharm. Anal.]. Unfortunately, the analytical part in its current form is not sufficient. Furthermore, the authors described KBA as major components. A recently published study showed that alpha and beta isomers of KBA cannot be separated and therefore selectively quantified by C18 phases [Schmiech et al., 2021, Molecules]. As KBA seems to be the main component, the authors should discuss this fact.

  • Antimicrobial and cytotoxic assay: The authors compared the zone inhibition of Fr-extract, Fr-SNEDS, and plain SNEDS (Table 6). Unfortunately, MBC and MIC were only determined for Fr-Extract and Fr-SNEDS and not for plain SNEDS. Why did the authors not determine this values for plain SNEDS? Noteworthy, the authors investigated the in vitro cytotoxic effects on A549 cells for Fr-Extract, Fr-SNEDS, and plain SNEDS. However, the cytotoxic effect should be further investigated on more lung cancer cell lines and on a non-cancerogenic control groups as the authors promote their study as “cytotoxic activities” within the title. Either further cell lines should be investigated or the title should be adjusted. Moreover, the title should clarify that the current study includes only in vitro investigations.

Minor comments:

  • Line 38. Please specify: …obtained from trees of the genus Boswellia,…
  • Line 42. Please specify: …and their 3-O-acetyl esters.

Reviewer 3 Report

In this work, the authors describe a new self-nanoemulsifying drug delivery system to enhance antimicrobial and cytotoxic activity of frankincense. The topic results interesting to the pharmaceutical development field and the research is well-conducted, however the way it is presented currently reduces the overall quality of the manuscript. Unfortunately, it is not suitable for publication in its current state, but if the authors address this comments hopefully the quality will be improved and the article will be published on Scentia Pharmaceutica.

  • I have mixed feelings about the fact of enhancing cytotoxic activity. I understand the authors want to engineer a formulation which would be suitable to treat both bacterial infections and cancer, but I have safety concers about that. It’s obvious that current chemotherapy lacks of specificity and unfortunately cancer patients still suffer from many undesired effects from their treatment, but would it be a good idea to treat infections with such a toxic compound? I mean, when using antimicrobial therapy “we want to kill the microorganism, not ourselves”.
  • This is the first time I see these systems named as “self-nanoemulsifying delivery systems/SNEDS”, I always have heard “self-nanoemulsifying drug delivery systems/SNEDDS”. Probably the authors could also utilise this term
  • The use of “formula” through the manuscript sounds a bit odd. I’d suggest using “formulation” or “composition” instead
  • Introduction, lines 46-52: I think this part should be rephrased. I can understand what the authors state, describing the activity of each compound, but it’s written in a quite lazy way. Just random sentences, I think this should be improved and linked to tell a whole story.
  • Line 54: verb is missing.
  • Overall, the way methods are reported needs to be improved. Here are several examples:
    • Line 71: it should be “design of experiments”. Please rephrase the sentence that states what DOE is, it is not clear.
    • Section 2.3.1: font size in the equation is too large, I can’t read it
    • Section 2.3.2: authors could report parameters of DLS, such as run time, or which parameter they choose for particle size (D50, particle size expressed in volume…)
    • Line 159: “using A ZetaSizer”.
    • Section 2.4.1: authors should state what DS and PDI are
    • Section 2.4.2: “using A UV-vis spectrophotometer”
    • Section 2.4.3: this section should be improved. Not clear how it was performed
    • Section 2.4.4: transmission electron microscopy instead of transmission electronic microscopy. In this section, authors could state different parameters of TEM, such as accelerating voltage
    • Section 2.5.1: it should be named “preparation of the inoculum”
    • Section 2.6: plate reader should be described (model of instrument, location). I’d suggest the equations would be reported in a different line, it is very difficult to read.
    • Section 2.7: “were optimized” and “in A dark place”. Drug content was not evaluated in the stability study? It’s important to verify physical stability, but I’m surprised chemical stability has not been reported.
    • A statistical analysis section needs to be included
  • Quality of the figures needs to be improved.
  • Section 3.1: if authors are reporting Rt and content in a table, I’m not sure if Figure 1 is needed.
  • Section 3.3.3: description of TEM images should be improved.
  • Section 3.4: what does “zone of inhibition” mean? Authors should state the meaning of MBC and MIC
  • Figure 7: combining bars and lines in the same graph looks confusing. I suggest the authors would split MBC and MIC in two different graphs
  • Section 3.5: a table or graph reporting the cytotoxicity values would be appreciated.
  • Section 3.6: this section should also be improved.
  • Discussion might also be improved. Looks a bit short taking into account the amount of experiments that had been performed. Also, I would start the discussion introducing the novelty of this work rather than a description of frankicense

Reviewer 4 Report

  Generally, the present paper is a well designed and well performed work that provides some novelty to the field of SNEDS for its application to a resin frankincense extract (Fr-extract) from the genus Boswellia, suitable to treat some lung infections and  lung adenocarcinoma.

  Major points:  

1) The quality of the images is not good enough. For example, fig 4 and 5 are blurry. The axis legends are not readable. Also in figure 5 it is repetitive to include the contour plot and the surface response chart. 

  -2) The optimised formulation contains ethanol as cosurfactant, do you believe it is the best excipient to use? The risk of using ethanol is its volatile behaviour which can alter the composition of the SNEDS over  time. Have you tested the stability of the formulation over time?. Did you have evaporation issues?.Also, the optimised particle size is below 20 nm, do you think this size is thermodynamically stable?

-4) Table 5 is not really needed in the text since all data are again compiled  in the text.  

-5.- Error bars and stats are missing in Figure 7.

-6.- The discussion needs some reinforcement in comparing and  contrasting your results to those from other authors.

What are the advantages of the SNED formulation developed by yourself in terms of chemical and pysical stability?. I recommend to read the paper: Repurposing butenafine as an oral nanomedicine for visceral leishmaniasis. Pharmaceutics 201911(7),53; https://doi.org/10.3390/pharmaceutics11070353P

Round 2

Reviewer 1 Report

The authors addressed the reviewer's concerns, therefore I would recommend the publication of the article

Reviewer 2 Report

Dear authors, thank you for revising the manuscript. On basis of your corrections and notations, I propose to “accept the manuscript in present form”.

However, please consider to improve your analytical (chemistry) effort in your next study. Especially, if you plan the publish in journals with higher IF.